# Ultrafast charge ordering by self-amplified exciton–phonon dynamics in TiSe$_2$

Chao Lian [1], Sheng-Jie Zhang [1], Shi-Qi Hu [1], Meng-Xue Guan [1] & Sheng Meng [1,2,3]*

The origin of charge density waves (CDWs) in TiSe$_2$ has long been debated, mainly due to the difficulties in identifying the timescales of the excitonic pairing and electron–phonon coupling (EPC). Without a time-resolved and microscopic mechanism, one has to assume simultaneous appearance of CDW and periodic lattice distortions (PLD). Here, we accomplish a complete separation of ultrafast exciton and PLD dynamics and unravel their interplay in our real-time time-dependent density functional theory simulations. We find that laser pulses knock off the exciton order and induce a homogeneous bonding–antibonding transition in the initial 20 fs, then the weakened electronic order triggers ionic movements antiparallel to the original PLD. The EPC comes into play after the initial 20 fs, and the two processes mutually amplify each other leading to a complete inversion of CDW ordering. The self-amplified dynamics reproduces the evolution of band structures in agreement with photo-emission experiments. Hence we resolve the key processes in the initial dynamics of CDWs that help elucidate the underlying mechanism.

[1] Beijing National Laboratory for Condensed Matter Physics and Institute of Physics, Chinese Academy of Sciences, Beijing 100190, P. R. China. [2] School of Physical Sciences, University of Chinese Academy of Sciences, Beijing 100190, P. R. China. [3] Songshan Lake Materials Laboratory, Dongguan, Guangdong 523808, P. R. China. *email: smeng@iphy.ac.cn

Charge density wave (CDW) in 1T-TiSe$_2$ has been one of the persistent eye-drawing topics over decades. It is not only an excellent playground to study the interplay between CDW and superconductivity[1–23], but also an evidenced excitonic insulator[24–28]. It was heavily debated which mechanism —the electron–phonon coupling (EPC)[29–40] or the excitonic pairing[40–61]—is the major driving force for the formation of CDW in TiSe$_2$. These two mechanisms disagree on the role of periodic lattice distortions (PLD) in CDW: PLD is essential in forming the CDW according to the EPC mechanism, while it is only a passive consequence of the CDW if the excitonic pairing dominates.

Isolating the PLD from the CDW can solve the debate, but it is not achievable in the ground state. Inspiringly, the ultrafast measurements can distinguish PLD and CDW based on their different time scales[62]. However, conclusions from previous ultrafast measurements are highly controversial. Utilizing time-resolved (tr) and angle-resolved photoemission spectroscopy (ARPES) measurements, Rohwer et al. observed very fast (<30 fs) collapses of the long-range order[63], which was interpreted by Mathias et al. as the signals of the mutually amplified carrier multiplication and gap quenching[64]. However, this mechanism completely neglects the possible involvements of ionic dynamics and the effects of geometry relaxation. Möhr-Vorobeva et al. observed the nonthermal melting within 250 fs in the ultrafast X-ray measurements and suggested that excitonic pairing generates the CDW[65]. Hellmann et al. also supported the excitonic mechanism[66]. They observed that the 100 fs change in the tr-ARPES signals is comparable to the buildup time of the electron-hole screening in the exciton formation. On the other hand, Porer et al. indicated that excitonic pairing was not the sole driving force of CDW[67]. They separately studied the electronic and structural orders via monitoring the characteristic peaks of the two in the transient energy loss spectra. They found that the PLD can persist with the quenched excitonic order. Despite extensive efforts in the past, the conclusions in all these previous studies were mainly derived from indirect mappings between the spectra and the CDW/ PLD orders. Notwithstanding the cutting-edge techniques used in these studies, the time-resolved spectra can only provide the averaged response of the material without atomic resolution. With recent significant progresses in real-time (rt) time dependent (TD) density functional theory (DFT) algorithms and computing power[68–81], it become now possible to perform ultrafast quantum dynamics simulations fully from first principles, to provide a unified atomic-level picture of ultrafast CDW dynamics.

Here we take advantages of recently developed rt-TDDFT methods for ab initio simulations of ultrafast dynamics in complex materials such as TiSe$_2$. We demonstrate that laser pulses knock down the CDW order without disturbing PLD through inducing a homogeneous bonding–antibonding electronic transition. The reduced CDW order then triggers the ionic movements exactly antiparallel to the original PLD, but cannot solely drives the observed inversion in CDW/PLD. Instead, assisted by EPC, a self-amplification mechanism between electron dynamics and lattice distortion emerges after the initial excitation, reproducing well experimental features observed in tr-ARPES measurements. We propose that both exciton pairing and EPC contribute to the CDW formation, albeit in a different timescale (see Fig. 1): CDW is predominantly initiated by exciton binding (<20 fs) and subsequently enhanced by EPC (>20 fs). These insights hint for a complete microscopic understanding on the nature of charge ordering in quantum materials.

## Results

**Ground state properties**. We calculate the atomic geometries of the normal phase and the CDW $2 \times 2 \times 2$ phase. The bulk TiSe$_2$ is used with the interlayer separation of 6.69 Å. The optimized PLD displacements $\{d_i\}$ are shown in Fig. 2a, where the displacements $\delta_{Ti} = 0.091$ Å, $\delta_{Se} = 0.030$ Å and $\delta_{Ti}/\delta_{Se} = 3.03:1$. These results excellently reproduce the experimental measurements $\delta_{Ti} = 0.085 \pm 0.014$ Å and $\delta_{Ti}/\delta_{Se} \sim 3:1$[82]. To directly compare with ARPES measurements, we calculate the effective band structure (EBS) along K-M-Γ-K by unfolding energy bands from the $2 \times 2$ BZ to the $1 \times 1$ BZ, as shown in Fig. 2. The calculated EBS agrees well with the experimental spectra. In the normal state, we reproduce the electron and hole pockets at M and Γ points, respectively. The CDW opens an indirect gap of 0.18 eV and creates folded valence bands (VB) at M and folded conduction bands (CB) at Γ. We note that the PBE+U approach with the Hubbard $U_{Ti} = 3.5$ eV incorrectly predicts that the CDW state is unstable[83]. Thus, we use $U_{Ti} = 0$ eV for all the dynamic simulations.

Based on the consistency between DFT results and experimental measurements, we briefly discuss the accuracy of DFT in describing the excitonic paring and EPC in TiSe$_2$. It is well known that the semilocal exchange-correlation (XC) functionals (e.g., PBE) poorly describes the long-range Coulomb screening $2\pi/|\mathbf{k}' - \mathbf{k}| \sim \infty$ in vertical excitations $\mathbf{q} = \mathbf{k}' - \mathbf{k} = 0$[84]. Computationally expensive corrections such as Bethe-Salpeter equation (BSE)[85,86] can considerably improve the accuracy. The experimentally observed superlinear feature is absent in our simulations (Supplementary Fig. 2). However, the long-range attractions produce spatially uniform forces on the ions. Based on the concept of the excitonic insulator[24–26], the PLD stability is only affected by the inter-valley excitons formed by an attractive interaction $V(\mathbf{w})$ between the electron pocket at the M point and the hole pocket at the Γ point, as shown in Fig. 2d. Here, $\mathbf{w} = \pm \mathbf{b}_i/2$ and $\mathbf{b}_i$ ($i = 1, 2$) being the reciprocal lattice vector along the $i$th direction. Therefore, $V(\mathbf{w})$ is a short-range interaction with a characteristic length scale $1/|\mathbf{w}| = a$, where $a = |\mathbf{a}_i|$ and $\mathbf{a}_i$ is the lattice vector. Note $V(\mathbf{w})$ is different from typical long-range interactions in

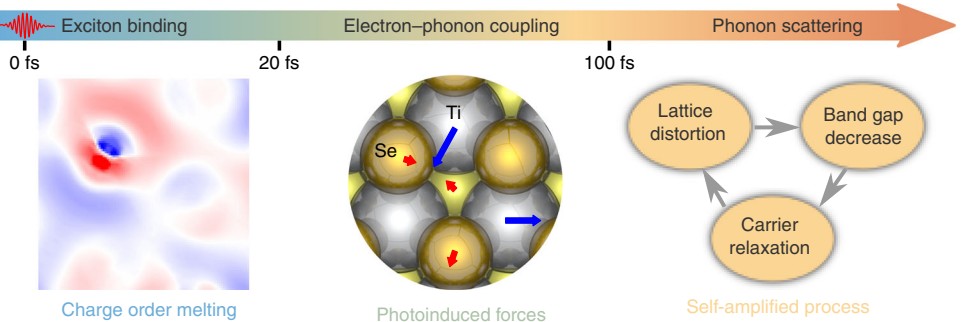

**Fig. 1 Schematic of atomic processes in photoexcited 1T-TiSe$_2$.** The laser pulse melts charge order within 20 fs, producing the forces that trigger the ionic movements. The self-amplified dynamics is assisted by electron–phonon couplings after initial excitation.

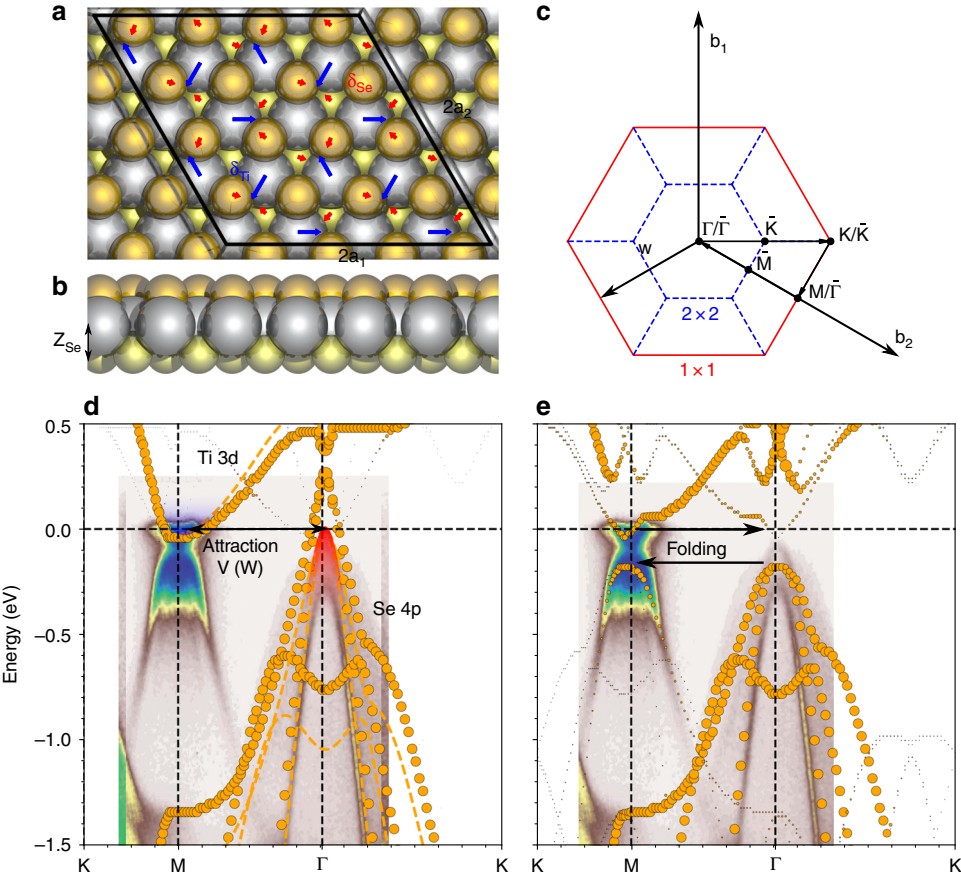

**Fig. 2 Atomic structure and electronic properties of 1T-TiSe$_2$. a** Top view and **b** side view of the atomic structure of 1T-TiSe$_2$. The silver, yellow and orange spheres denote the Ti atoms, the Se atoms on the bottom layer and the Se atoms on the top layer, respectively. The blue and red arrows denote the PLD displacements $\{d_i\}$ of the Ti and Se atoms, respectively. **c** Brillouin zones (BZ) of TiSe$_2$. The solid red lines and blue dash lines denote the BZ of the $2 \times 2$ and $1 \times 1$ cell, respectively. $\{\Gamma, M, K\}$ and $\{\bar{\Gamma}, \bar{M}, \bar{K}\}$ denote the special $k$ points in $1 \times 1$ and $2 \times 2$ BZ, respectively. Effective band structures (EBS) of the **d** normal state with $U = 0$ eV (dots) and $U = 3.5$ eV (dashed lines) and **e** CDW state. The plots in shaded areas in (**d**) and (**e**) are ARPES spectra reproduced from ref. [63]. The Fermi level is shifted to the experimental value.

Wannier excitons. The semilocal XC already includes exciton binding between the electron and hole pockets at M and $\Gamma$, respectively, albeit slightly underestimating the screening effect.

To quantitatively demonstrate the validity of the semilocal functional in describing the intervalley exciton with momentum $\mathbf{q} = \mathbf{w}$, we compare the linear-response TDDFT results obtained from adiabatic PBE (APBE) and BSE kernels. At $\mathbf{q} = \mathbf{w}$, the APBE kernel yields similar absorption spectra with those from BSE kernel (Supplementary Fig. 3). Since the BSE kernel is a well-accepted accurate description of excitons[85,86], this indicates that the electron-hole exchange effect has been well described in the semilocal XC. Therefore, due to the unique band structures of TiSe$_2$, the semilocal XC yields acceptable excitonic interactions.

Furthermore, the density functional perturbation theory and molecular dynamics calculations reproduce well experimental phonon spectra[87,88] and thermal conductivity[83,89], confirming that DFT can accurately describe the EPC in TiSe$_2$. Thus, the semilocal XC and electron-ion dynamics simulations are suitable for tracking photoexcitation physics of TiSe$_2$.

**Dynamics of charge orders**. To identify the dominant mechanism, we try to supply direct proofs by separating the CDW dynamics from the PLD in the time domain. Instead of directly creating CDW at the normal state, we aim at decreasing the CDW order with the fixed

PLD. We analyze the differential charge density $\rho_{tot}(\mathbf{r}, t) = \rho_{chg}(\mathbf{r}, t) - \rho_{atom}(\mathbf{r}, t)$ as a function of time, where $\rho_{chg}$ is the charge density and $\rho_{atom}$ is the superposition of the atomic charge densities. Thus, $\rho_{tot}(\mathbf{r}, t)$ features the spacial distribution of the bonding $(+)$ and antibonding $(-)$ densities. We divide $\rho_{tot}(\mathbf{r}, t)$ into two parts $\rho_{1 \times 1}(\mathbf{r})$ and $\rho_{CDW}(\mathbf{r})$, where $\rho_{1 \times 1}(\mathbf{r}) = [\rho_{tot}(\mathbf{r} + \mathbf{a}_i) + \rho_{tot}(\mathbf{r})]/2$, and $\rho_{CDW}(\mathbf{r}) = \rho_{tot}(\mathbf{r}) - \rho_{1 \times 1}(\mathbf{r})$. Obviously, $\rho_{1 \times 1}(\mathbf{r} + \mathbf{a}_i) = \rho_{1 \times 1}(\mathbf{r})$ characterizes the original $1 \times 1$ order, while $\rho_{CDW}(\mathbf{r})$ characterizes the strength of the CDW order. A laser-induced charge difference is characterized by $\Delta \rho_i(\mathbf{r}) = \rho_i(\mathbf{r}, t_f) - \rho_i(\mathbf{r}, t_0)$ ($i = $ tot, $1 \times 1$, CDW). Figure 3 shows the two-dimensional contour of the charge density $\rho_i(x, y, t)$ ($i = $ tot, $1 \times 1$, CDW), which is $\rho_i(\mathbf{r}, t)$ averaged over $z$ direction. Comparing Fig. 3a–c, we find that $\rho_{1 \times 1}(x, y, t_0)$ is the main ingredient of $\rho_{tot}(x, y, t_0)$, even with the presence of PLD, while $\rho_{CDW}(x, y, t_0)$ is localized around the Ti positions. Starting from the CDW ground state with the fixed PLD, the CDW order $\rho_{CDW}(x, y, t)$ decreases after the laser illumination: (i) The laser induces electron transfer from the bonding area to the antibonding area. The induced charge $\Delta \rho_{tot}(x, y, t)$ (Fig. 3g) is opposite to ground state charge $\rho_{tot}(x, y, t_0)$ (Fig. 3a). (ii) The majority of the induced charge has the $1 \times 1$ periodicity, i.e., $\Delta \rho_{1 \times 1}(x, y, t)$ (Fig. 3h) dominates $\Delta \rho_{tot}(x, y, t)$ (Fig. 3g). (iii) The most important feature is that the induced CDW charge density $\Delta \rho_{CDW}(x, y)$ (Fig. 3i) is opposite to the original $\rho_{CDW}(x, y, t_0)$ (Fig. 3c), indicating a 20% decrease in the excitonic order.

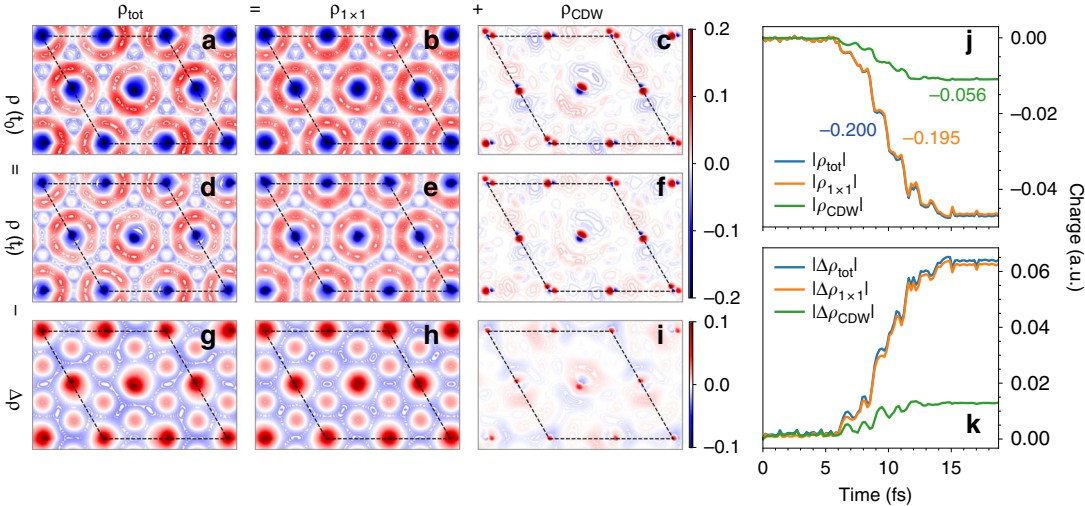

**Fig. 3 Dynamics of charge order**. Two-dimensional contour plots for **a** $\rho_{tot}(x, y, t_0)$, **b** $\rho_{1\times1}(x, y, t_0)$, **c** $\rho_{CDW}(x, y, t_0)$, **d** $\rho_{tot}(x, y, t_f)$, **e** $\rho_{1\times1}(x, y, t_f)$, **f** $\rho_{CDW}(x, y, t_f)$, **g** $\Delta\rho_{tot}(x, y)$, **h** $\Delta\rho_{1\times1}(x, y)$, **i** $\Delta\rho_{CDW}(x, y)$, where $t_0 = 0$ and $t_f = 20$ fs. **j** The bonding charge and **k** anti-bonding charge as a function of time. The initial values in (**j**) are shifted $-0.2$, $-0.195$, $-0.056$ for $\rho_{tot}$, $\rho_{1\times1}$, $\rho_{CDW}$, respectively.

We analyze the integrated charges $Q_i(t) = \int |\rho_i(\mathbf{r}, t)|d\mathbf{r}$ and $C_i(t) = \int |\rho_i(\mathbf{r}, t) - \rho_i(\mathbf{r}, t_0)|d\mathbf{r}$ ($i = $ tot, $1\times1$, CDW) for quantitative comparisons. The former characterizes the strength of bonding, while the latter denotes the weakening of bonding states, i.e., the strength of anti-bonding. As shown in Fig. 3j, the percentage of the decrease $[Q_i(t_f) - Q_i(t_0)]/Q_i(t_0)$ is 23.6%, 23.8% and 19.7% for $i = $ tot, $1\times1$, CDW, respectively. Meanwhile, $C_i(t)$ increases, with a ratio $C_{tot}(t_f):C_{1\times1}(t_f):C_{CDW}(t_f) = 1:0.98:0.20$, slightly different from the ratio of the initial bonding charges $Q_{tot}(t_0):Q_{1\times1}(t_0):Q_{CDW}(t_0) = 1:0.98:0.27$. Thus, laser-induced bonding–antibonding transfer is nearly homogeneous, lowering both the $1\times1$ and CDW order proportionally. Since the decrease in $|Q_{1\times1,bond}|$ affects all chemical bonds homogeneously, as an overall effect, the decrease in $|Q_{tot,bond}|$ lowers the stability of CDW.

**Photoexcited lattice dynamics**. The laser-induced CDW instability would further trigger dynamic changes in PLD, which are simulated from first principles. Among the atoms in the layer of $2\times2$ cell, only one Ti atom and one Se atom are nontrivial symmetry-inequivalent atoms, as shown in Fig. 4a. We present the time-dependent displacement $\mathbf{d}_i(t)(i = \mathrm{Ti, Se})$ by the projection of $\mathbf{d}_i(t)$ on the PLD direction $\delta_i(t) = \mathbf{d}_i(t) \cdot \mathbf{d}_i(t_0)/|\mathbf{d}_i(t_0)|$ and the angle $\theta(t)$ between $\mathbf{d}_i(t)$ and $\mathbf{d}_i(t_0)$ (Fig. 4b). We find that laser triggers a set of movements $\Delta\mathbf{d}_i(t) = \mathbf{d}_i(t) - \mathbf{d}_i(t_0)$ which are exactly antiparallel to $\mathbf{d}_i(t_0)$, noticing that $\delta_i$ decreases with all $\cos\theta \approx \pm1$ as shown in Fig. 4d–e. Besides the in-plane movements, the overall bonding–antibonding charge transition introduces an out-of-plane breathing mode between the Se layer and the Ti layer $z_{Se}(t)$ (Fig. 4f). This is the established out-of-plane $A_{1g}$ mode with the periodicity of 175 fs (5.7 THz) when $I = I_0/4$, which is slightly smaller than the experimental value 5.9 THz[90,91]. Higher laser fluence $I = I_0$ further decreases the frequency to 5.1 THz, due to the weakened Ti–Se bonds by photocarriers.

We analyze the dynamical potential energy surface (PES) related to the PLD dynamics. To focus on the in-plane PLD movements other than the out-of-plane $A_{1g}$ mode, we plot the minus of the kinetic energies of these two movements $-E_{kin}$ and $-E_{kout}$, respectively. Since the total energy $E_{tot}(t) = E_p(t) + E_{kin}(t) + E_{kout}(t)$ is conserved, where $E_p(t)$ is the potential energy, we use $-E_{kin}(t)$ as a function of $\delta(t)$ to characterize the

dynamical PES of the PLD, as shown in Fig. 4c. The dynamical potential energy decreases due to the in-plane PLD movements, together with the energy oscillations attributed to the out-of-plane $A_{1g}$ mode (Fig. 4g). This indicates that the laser induced PLD movements stabilize the system. Within the simulation time, $\mathbf{d}_i(t)$ has been completely inverted $\mathbf{d}_i(t) = -\mathbf{d}_i(t_0)$ at around 300 fs, lowering the dynamical PES and creating a new quasi-equilibrium state. We note that the inversion of PLD produces a degenerate CDW state which is symmetry-equivalent to the original state.

The above data imply a scenario of ultrafast PLD switch, which includes three consecutive steps: (i) In the first 20 fs, the laser pulses pump the electrons from the bonding state to the anti-bonding state, which leads to the change of the PES. The calculated double-well PES yields a vibration frequency of 4.3 THz, which is consistent with the oscillation period of the amplitude mode of about 220–240 fs observed in dynamic simulations (Fig. 4d). This frequency is comparable to the experimental value of 3.4 THz observed for the $A_{1g}$ CDW amplitude mode[90,91]. The difference between theory and experiment is attributed to possible mixing of other phonon modes and inadequate accuracy of semilocal functionals to treat low-energy phonons. The softening of the double-well potential is evidenced by the fact that the CDW mode oscillations disappear when the laser fluence increases from $I_0/4$ to $I_0$. (ii) The lattice transforms into a structure with the opposite PLD along the non-equilibrium TD-PES during 20–300 fs. (iii) The photoexcited system might relax into the ground state after the recombination of photocarriers. This process is beyond our simulations and requires multichannel and large-scale modeling to properly account for the decoherence[92–94] and dissipation effects[95,96].

We complete the story by briefly discussing the long-time behaviors. In the experiments, the ionic movements in different unit cells have different phases due to the finite thickness of the material, the inhomogeneous spatial distribution of the laser spot, as well as thermal fluctuations. Via phonon–phonon scattering, the equilibrium temperature gradually forms at the timescale of picoseconds. The thermally driven phase transitions may occur through an increasing population of phonons and changes in the atomic potential via anharmonic phonon interactions. Thus, the nonthermal CDW dynamics observed here is separated from thermal transitions at different time scales.

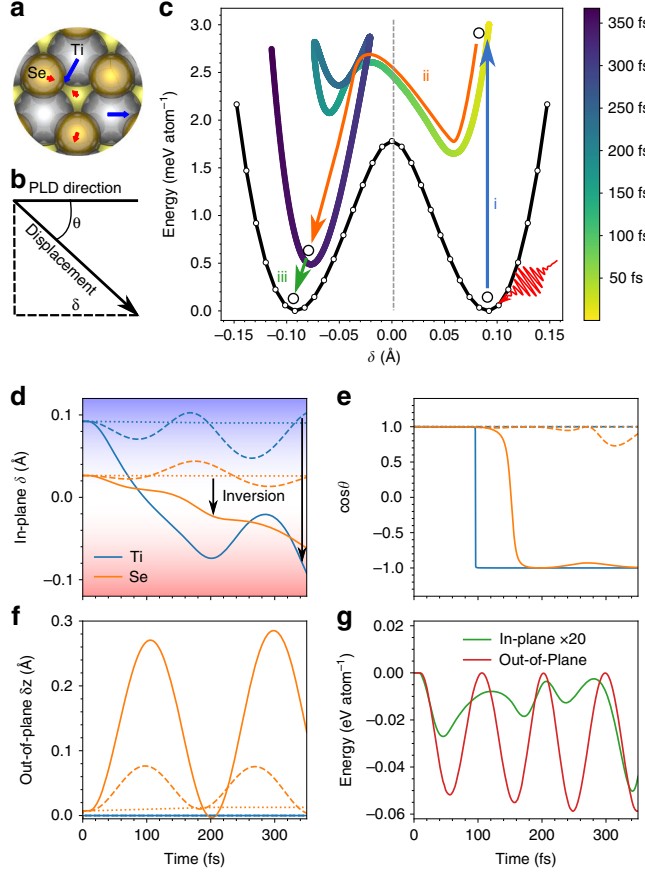

**Fig. 4 Photoexcited lattice dynamics. a** All the symmetrically inequivalent atoms with nontrivial PLD. **b** The illustration of $\delta$ and $\theta$. **c** The potential energy surfaces (PES). The black and colored lines denote the ground state PES and the non-equilibrium TD-PES, $E[\delta(t)]$, respectively. The **d** $\delta(t)$, **e** $\cos\theta(t)$, and **f** $\delta z(t)$ as a function of time. **g** The minus of the kinetic energies of out-of-plane movements $-E_{\mathrm{kin}}(t)$ and in-plane movements $-E_{\mathrm{kout}}(t)$ as a function of time. The solid, dash and dotted lines denote the $I = I_0$, $I = I_0/4$, and the quenched case ($I = I_0$ but with 1.4% energy dissipation), respectively.

**Dynamic interplay**. The highly directional PLD movements are clearly not the reason but the consequence of the reduced excitonic order. In contrast to the complete PLD inversion for the $I = I_0$ case, $\mathbf{d}_i(t)$ only oscillates around the original value when laser intensity decreases to $I_0/4$. The formation/decay of the CDW/PLD is electronically initialized, which supports the excitonic mechanism. This intensity dependence is also experimentally observed[65,67]: the $A_{1g}$ CDW mode is softened as the laser fluence increases, which serves as a precursor of the PLD melting. Meanwhile, there are extra factors in the following dynamics. Assuming that the excitonic pairing is the only driving force, an inversion of electronic order $\rho_{\mathrm{CDW}}(\mathbf{r}, t_f) = -\rho_{\mathrm{CDW}}(\mathbf{r}, t_0)$ is required to reverse the PLD $\mathbf{d}_i(t) = -\mathbf{d}_i(t_0)$. Considering $\rho_{\mathrm{CDW}}$ only decreases by about 20%, the PLD should be only perturbed.

We design another numerical experiment by introducing a thermostat and removing a small fraction of kinetic energy at the rate of 0.01 eV atom$^{-1}$ ps$^{-1}$. We note that the kinetic energy $E_k(t)$ increases by 0.66 eV atom$^{-1}$ ps$^{-1}$ within 60 fs, as shown in Fig. 4g. Thus the thermostat, affecting only 1.4% of the kinetic energy change, is expected to slightly delay the dynamics of PLD $\delta(t)$. However, despite that the dissipation in the kinetic energy is small

(1.4%), the PLD dynamics is completely quenched (Fig. 4, dotted lines). It strongly implies that the increase in $E_k(t)$ is nonlinear, with a lower rate at the beginning. The fragile initial movement of $\delta(t)$ will be completely quenched with a small dissipation rate. Since the PLD dynamics is sensitively dependent on its own trajectory, we infer that there exists a self-amplified electron–phonon mechanism in the PLD dynamics initiated by exciton binding.

Accordingly, we demonstrate the impact of ionic movements on electronic structure through analyzing TD-EBS along Γ-M in Fig. 5a–f. The laser pulse first shifts the EBS downwards and enlarges the bandgap; then VBs are raised by ~0.18 eV while CBs barely change during 20-80 fs, in excellent agreement with tr-ARPES data. It leads to the vanishing band gap and forms hole and electron pockets, boosting the relaxation of hole carriers towards the maximum of VBs (VBM). The carrier relaxation lowers the total energy of the system. In comparison, in the quenched case where ions barely move (Fig. 5h–l), the dynamic electronic evolution—the gap closing and carrier relaxation—are completely absent, implying that the measured changes in EBS are highly correlated with ionic movements. We thus identify a self-amplified process from these observations: the decrease in the PLD → the decrease in the bandgap → the decrease in the energy of carriers → the further decrease in PLD. We expect such a dynamic interplay between electronic order and PLD, ignored in previous studies, would sustain in a variety of CDW materials such as NbSe$_2$, TaS$_2$, LaTe$_3$, etc.[89,97] The results highlight the entangled dynamics between different degree of freedoms in quantum materials.

## Discussion

To complete our story, we note the following facts towards building a unified picture for experimental observations.

(i) The effective lattice temperature is relatively low during the whole laser-induced dynamics. The maximum kinetic energy of ions is 0.06 eV atom$^{-1}$ in a temporary period (corresponding to a transient effective temperature of 450 K). Although the low kinetic energy of ions may not be directly connected to nonthermal processes, we believe the processes discussed above are primarily nonthermal since they are short-lived (<300 fs) and the excited-state carriers are clearly present (Fig. 5). In this case the structural soft-mode potential is relaxed or even driven to a single well-potential by excitation of the electronic degree of freedom. This is also consistent with the experimental observations[65] and previous studies[97–99].

(ii) The CDW order is disturbed instantly by the laser pulse within the timescale of 10 fs, since it is originated from homogeneous electronic transitions. No buildup time in electron-hole screening is needed. It shows that, the ~100 fs shift in tr-ARPES signals[66] is not caused by the buildup of electron-hole screening, but by the EPC-assisted PLD dynamics $\mathbf{d}_i(t)$. Apparently, the phonon response within 100 fs is contradictory to the well-accepted timescale of phonon dynamics $\tau_{\mathrm{ph}} > 1$ ps. However, we note that the photo-induced PLD dynamics are coherently driven not by the phonon–phonon scattering, but directly by the electronic transitions. Thus, the response shares the same timescale as the electron–phonon scattering of ~100 fs.

(iii) With the laser intensity used in our simulations, the laser pulses induce the PLD dynamics but barely change the intrinsic phonon modes (e.g., out-of-plane $A_{1g}$). This explains the sustainable characteristic peaks of PLD with the melted exciton order as previously observed in ref. [67].

(iv) There are open discussions in literature on whether excitonic interaction or the Jahn-Teller distortion is dominant for the formation of CDW in TiSe$_2$. Our simulations seem to indicate the two effects could coexist in a short period perturbed by

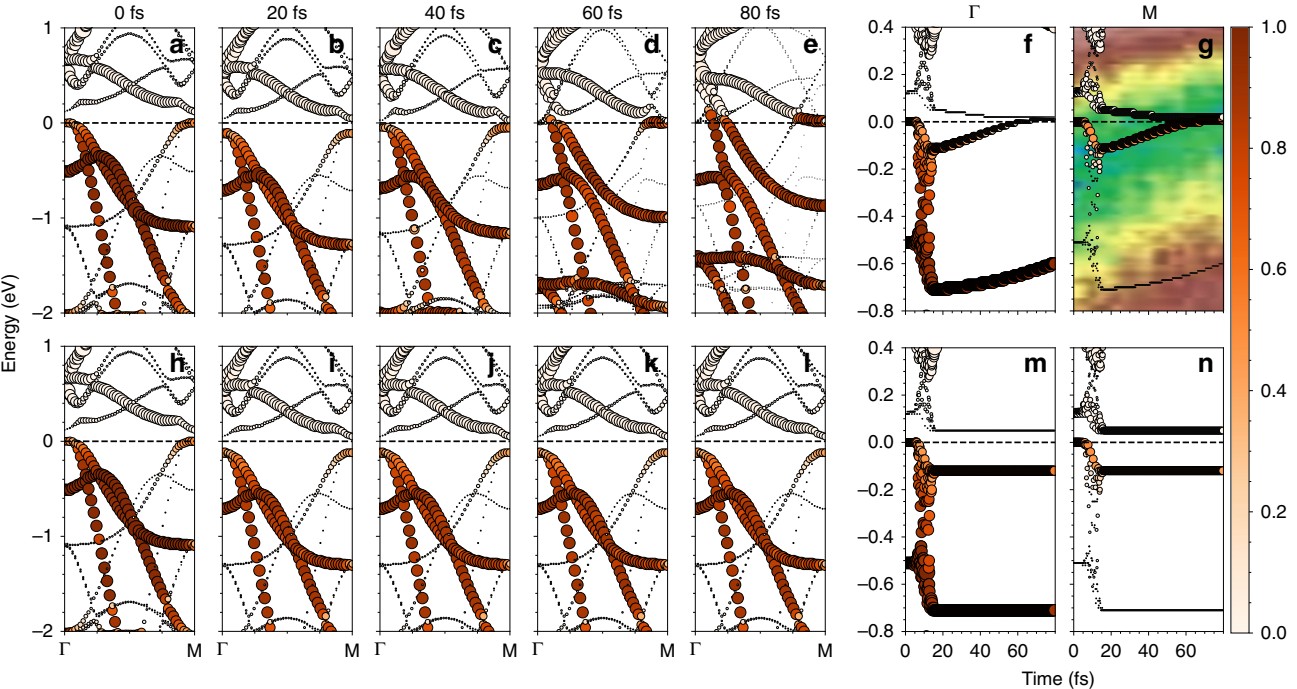

**Fig. 5 Time dependent band structure.** Snapshots of TD-EBS for (**a–e**) the PLD dynamics, and (**h–l**) the quenched case. EBS at $\Gamma$ and M point as a function of time for (**f–g**) PLD dynamics and (**m-n**) the quenched case. The color bar denotes the carrier population. The squares in (**g**) mark the experimental tr-ARPES data, reproduced from ref. [66].

photoexcitation. From Figs. 3 and 4 it is shown that while there is no significant lattice distortion (thus no changes in Jahn-Teller interaction), the electronic order is suppressed by 20% in the first 20 fs, beaconing excitonic interactions. On the other hand, the following self-amplifying process suggests dynamic Jahn-Teller lattice distortion is crucial for the subsequent decay/formation of CDW state.

We demonstrate an entangled exciton-phonon mechanism for charge ordering in 1T-TiSe$_2$ by ab initio ultrafast rt-TDDFT simulations of photoexcitation dynamics. We show that laser pulses knock down the CDW order via inducing a homogeneous bonding–antibonding electronic transition. The weakened CDW order then triggers ionic movements exactly anti-parallel to the original PLD, but cannot solely drive the observed PLD inversion. Instead, assisted by the EPC, a self-amplification mechanism drives the PLD dynamics initiated by laser excitation, in excellent agreement with the tr-ARPES measurements. We propose that both the excitonic pairing and the EPC contribute to the CDW/PLD formation, but in a different timescale: the excitonic pairing initializes the formation of CDW, while the EPC promotes the following dynamics through a self-amplification process.

## Methods
**Numerical calculations.** Following our previous efforts in developing time dependent ab initio package (TDAP)[100–102], we implemented TDDFT algorithm in the plane-wave code `Quantum Espresso`[103]. The details are described in Supplementary Note I. We used the projector augmented-waves method (PAW)[104] and the Perdew-Burke-Ernzerhof (PBE) exchange-correlation (XC) functional[105] in both DFT and TDDFT calculations. Pseudopotentials were generated using `pslibrary`[106]. The plane-wave energy cutoff was set to 55 Ry. Brillouin zone (BZ) was sampled using Monkhorst-Pack scheme[107] with a $6 \times 6 \times 3$ $k$-point mesh. Band unfolding techniques were utilized to generate the EBS[108,109] with a modified version of `BandUP` code[110–112]. Onsite Coulomb repulsion $U = 3.5$ eV was added to the Ti atom to reproduce the experimental band structure, while we used $U = 0$ in the dynamic TDDFT calculations and structural optimization. The electron timestep $\delta t$ is $4 \times 10^{-4}$ a.u. $= 0.1935$ attosecond and the ion timestep $\Delta t$ is 0.194 fs. Laser pulses with a wavelength $\lambda = 800$ nm, width $\sigma = 12$ fs and fluence $I_0 = 2.1$ mJ cm$^{-2}$ centered at $t = 10$ fs is utilized. The linear-response TDDFT calculations were carried out with the YAMBO package[113]. This setup, which is similar to that used in experiment, reproduces the measured number of excited carriers in TiSe$_2$ (Supplementary Fig. 2).

## Data availability

The data that support the findings of this study are available from the corresponding author upon reasonable request.

## Code availability

The code that was used to simulate the findings of this study is available from the corresponding authors upon reasonable request.

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

## Acknowledgements

We acknowledge partial financial supports from MOST (grants 2016YFA0300902 and 2015CB921001), NSFC (grants 91850120, 11774396, 11934003), and CAS (XDB07030100).

## Author contributions

S.M. conceived and directed the research. C.L. performed the calculations and analyzed the data. C.L., S.-J.Z., S.-Q.H., M.-X.G. and S.M. participated in the discussion. C.L. and S.M. wrote the paper with contributions from all the other authors.

## Competing interests

The authors declare no competing interests.
