## [Peer Review File · Nature Communications]

Reviewers' comments:

Reviewer #1 (Remarks to the Author):

This paper aims at explaining the microscopic origin of charge density wave (CDW) formation in TiSe₂.

It has long been debated whether CDW in TiSe₂ is driven by electron-phonon coupling or exciton pairing and this is testified by the large body of literature on the topic. Since identifying the dominant mechanism in real materials is a difficult task because electron-phonon and electron-electron interactions are often equally strong a way to come around the problem is provided, as presented in Ref [S. Hellmann, et al. Nat Comms 3, 1069 (2012)], by observing the time evolution of the CDW phase melting subsequent to an impulsive laser excitation.

In order to study the CDW melting the authors perform ab-initio real-time TDDFT simulations of the complete process including laser excitation, electron, and ions dynamics to find that both electron-electron interaction and electron-phonon interactions contribute to the CDW formation but on different time scales.

Though the time scale separation between the competing mechanisms is not a surprising result (since electron and phonon time scales are naturally well separated) it is remarkable how the whole process seems to be captured with great accuracy by TDDFT and the fact that the results fit well the experiments. The authors provide an extensive analysis of the combined electron-ion dynamics and explain how to isolate the different dominant mechanism in an otherwise complex intertwining between different degrees of freedom. Although the overall presentation is clear and adequate for a Nature journal some of the evidence for the finding in the paper is staked up in a way that is not always easy to follow (see comments below).

The topic is of interesting and the paper has the potential to represent a theoretical breakthrough to our current understanding of the problem.

However, I have two major points that could undermine the validity of the thesis presented that the authors should address:

1. It is well known that, in order to describe excitons with DFT/TDDFT, one needs to choose a functional capable to capture the long range electron-hole interaction and the screening. This is typically achieved by hybrid functionals (e.g. HSE) which contain a fraction of exact-exchange. The authors motivate their choice of a semi-local functional (PBE) by the observation that the system in the CDW phase presents a indirect gap and therefore the screening, which is proportional to $1/q$, is never divergent because the momentum transfer, q , connecting valence and conduction pockets is finite. While this observation is justified for the CDW phase in equilibrium in general this is not the case, in particular during the CDW melting process the system can find itself into direct-gap configuration for which the finite q argument is no longer valid and the xc functional is inadequate. How the authors justify their approximation in this situation?

2. How the experimental ARPES data of fig 5 has been obtained? In the paper it reminds the reader to Ref [65]. However the only figure in the reference that could contain data compatible with the figure is [65] fig. 3 which is a density plot that, at first sight, presents no clear connection with the data points reported in fig 5. The figure in [65] has a much larger time scale up to 2000 fs and the density is so broad that at the very least the presentation of the data in the paper without error bars is misleading.

Since this comparison constitutes the principal validation of the theory on the CDW melting it is crucial that the authors provide a clear account for the origin of the data.

In addition I have other minor points that the authors should consider:

3. From the text it not clear whether the theoretical result are obtained from a monolayer TiSe₂ or bulk. For instance the data in Fig. 2 present a comparison with experiments on monolayer while Fig.3 compares to bulk data.

4. It is remarkable the fact that the simulations, without any prior knowledge, reproduces the excitation of the out-of-plane A_{1g} phonon mode observed in IR and EELS spectra. The authors find a mode frequency of 4.5 THz while in the literature the value is 3.4 THz (and not 4 THz as reported in the paper)[M. Porer, et al. Nat. Mat. 13, 857 (2014)].

Can the authors comment on the origin of the mode stiffening predicted by TDDFT?

5. In order to highlight the role of the electron-phonon coupling in the lattice distortion the authors perform a simulation with a thermostat that keeps the ion energy fixed. How is the the thermostat implemented? Can the authors provide a more details and/or a reference?

6. The meaning of the sentence "Since the PLD dynamics is self-dependent, we infer that there exists a self-amplified electron-phonon mechanism in the PLD dynamics initiated by exciton binding" is a bit obscure. What does it means that the PLD dynamics is "self-dependent"? Can the authors rephrase to clarify the concept?

7. In Fig. S2 the authors present the photoinduced carrier density. Can the authors provide details how this quantity is calculated?

It is known that simply projecting the time dependent KS state $\psi_{\mathbf{k}}(t)$ on the the conduction bands at $t=0$ and same \mathbf{k} is not gauge independent and one should project the to equilibrium states at momentum $\mathbf{k}'=\mathbf{k}-\mathbf{A}(t)/c$ - see [T. Ohtobe, et al., Phys Rev B 77, 165104 (2008)] and [S. A. Sato and K. Yabana, Phys Rev B 89, 224305 (2014)]. Failing to do so provides artificial oscillations in the carrier density as a function of the the vector potential $\mathbf{A}(t)$ like the ones observed in the first panel of the figure.

8. A remark about point iii) of the PLD dynamics (page 4 left column line 7 from the bottom): "The photoexcited system might relax into the ground state after the recombination of photocarriers, which occurs beyond our simulation period". While I agree that this would be the expected physical long time behavior I think recombination will never be observable in the numerical model presented in this paper because of the lack of dissipation channels like e.g. electron-electron scattering, electron-phonon scattering, electron-defect scattering etc.

Reviewer #2 (Remarks to the Author):

Lian et al. present a time-dependent DFT study of CDW ordered TiSe₂. The authors simulate a scenario of femtosecond charge-transfer photoexcitation with high excess energy with parameters adapted to recent experimental studies.

In their manuscript, the authors report a fast suppression of the charge order on a timescale of 20 fs subsequent to a coherent charge-transfer excitation via an ultrafast field-pulse. The excitation process causes a population of antibonding states that leads to a reduced ordering of the electronic degree of freedom on a timescale of 20 fs. This relaxes the lattice distortion associated with the CDW. The latter process is found to amplify the reduction of the electronic charge ordering ('self-amplification').

The questions addressed in this paper (in particular the role of the crystal lattice in a self-amplified quench of a correlation induced ordering phenomenon) are of interest in the wider field of correlated electron systems and their dynamics on ultrafast timescales.

However, I do not recommend publication of the paper in the present state.

The main issue I see is the insufficient description of how the authors determine the structural dynamics from their calculations and how the DFT results are compared with experimental data.

Before publication, the authors need to clarify following issues:

- In their DFT calculation, they set a on-site Coulomb repulsion of the Ti sites of $U_{\text{Ti}}=3.5$ eV. It has been shown earlier [Bianco et al., PRB 92 094107 (2015)] that this approach indeed yields a good fit between the resulting DFT band structure and ARPES data, but it suppresses the CDW instability. Can the authors show their actual potential energy surface before the field pulse used in their time-dependent calculation sets in (i.e. not only show a schematic picture in Fig. 4(c))? Why is it different to that obtained by Bianco et al.? For low excitation they find an oscillation period of the amplitude mode of about 220 fs (~ 4.5 THz). How does this match with a double well potential for the CDW amplitude mode that is expected to strongly soften (or even become a single well-potential) due to a rather high value of U_{Ti} ?
- It is unclear to me from the manuscript how the structural motions are obtained from the potential. What is the equation of motion and which effective mass is used for the modes? How are the mode frequencies in Fig. 4 (CDW amplitude mode, out-of-plane mode) obtained?
- How is the dissipation of electronic excess energy or carrier cooling implemented (Fig 5 (a-e))? Is the electronic system only cooled via manual dissipation of energy solely via the coupled A1g coherent mode?

Further comments:

- The authors should elaborate why they claim that their results support excitonic interactions. In my opinion the results could equally well be interpreted in terms of the Jahn-Teller effect.
- Fig. 2 (d) does not show the meaning of $V(w)$ as written in the text.
- I am wondering to which extent the link between a 'nonthermal' phase transition (ref. 64) and the kinetic energy of lattice sites during the ultrafast transition is justified. Nonthermal in the sense of ref. 64 refers to the driving mechanism, meaning that the structural soft-mode potential is relaxed or driven to a single well-potential by excitation of the electronic degree of freedom (i.e. direct coupling of the structural potential to the electronic state). For a thermally driven phase transition, an increasing population of the vibrational system (e.g. via e-ph scattering after laser excitation) would change the structural potential via anharmonic phonon interactions. In this case the kinetic energy of the respective ions would be even lower as the structural dynamics induced by a changing potential is limited by the timescale of e-ph heat exchange.

I recommend an extended revision of the manuscript that includes addressing the above points. Furthermore, I suggest publication in a more specialised journal as the manuscript reports results obtained by an established computational technique for a specific class of materials.

Reviewer #3 (Remarks to the Author):

The authors present a real-time TDDFT study of photoexcited carrier dynamics in the material 1T-TiSe₂, investigating directly from first-principles, the interplay between charge-density-wave (CDW) order and periodic lattice distortions (PLD) in the time-domain. Starting from a ground state where CDW order and PLD co-exist, the authors show that photoexcitation reduces CDW order at early times (< 20 fs) resulting in excited-state forces that in turn drive a switching of PLD through nonadiabatic electron-ion dynamics on the 20-300 fs timescale. The simulations in this

work certainly present a detailed atomistic picture of photoexcitation dynamics in 1T-TiSe₂ and a plausible interpretation of some time-resolved ARPES experiments published previously. The methods employed by the authors namely real-time TDDFT combined with Ehrenfest dynamics are state-of-the art in a condensed-matter context. The simulations are also complemented by additional post-processing analysis which serve to clarify the detailed dynamics while making contact with ultrafast experiments.

Overall this is a detailed and impressive set of simulations and I would recommend publication after the authors have addressed the concerns/questions below:

(1) On page 2, the authors state: "Thus, the semilocal XC and electron-ion dynamics simulations are suitable for tracking photoexcitation physics of TiSe₂."

This statement needs some qualification. The argument preceding the above statement only justifies the use of semi-local functionals for an excitonic coupling that is short-ranged in real-space. It is not clear whether all aspects of photoexcitation including direct inter-band excitations and related excitonic effects are adequately captured at the level of semi-local functionals. Indeed in Fig. S2, the authors show that experimental values for photoexcitation differ from the simulations at intermediate intensities and attribute the difference to excitonic effects.

(2) On page 4, the authors state: "In contrast to the complete PLD inversion for the $I = I_0$ case, $d_i(t)$ only oscillates around the original value when laser intensity decreases to $I_0/4$."

What is the reason for this type of intensity dependence? This suggests that the amplitude of the ionic motions perhaps depends on the density of excited carriers rather than on the energy of excitation. Is this an artifact of employing the Ehrenfest approximation in calculating nuclear forces? Is there experimental evidence indicating that the PLD switch would exhibit a fluence dependence as the simulations suggest?

(3) The Methodology section should provide details of the type of electron-ion dynamics employed in the study including parameters related to the thermostat used and relevant literature citations in this regard.

Responses to reviewers' comments

Reviewer 1

Comments: *This paper aims at explaining the microscopic origin of charge density wave (CDW) formation in TiSe_2 .*

It has long been debated whether CDW in TiSe_2 is driven by electron-phonon coupling or exciton pairing and this is testified by the large body of literature on the topic. Since identifying the dominant mechanism in real materials is a difficult task because electron-phonon and electron-electron interactions are often equally strong a way to come around the problem is provided, as presented in Ref [S. Hellmann, et al. Nat Comms 3, 1069 (2012)], by observing the time evolution of the CDW phase melting subsequent to an impulsive laser excitation.

In order to study the CDW melting the authors perform ab-initio real-time TDDFT simulations of the complete process including laser excitation, electron, and ions dynamics to find that both electron-electron interaction and electron-phonon interactions contribute to the CDW formation but on different time scales.

Though the time scale separation between the competing mechanisms is not a surprising result (since electron and phonon time scales are naturally well separated) it is remarkable how the whole process seems to be captured with great accuracy by TDDFT and the fact that the results fit well the experiments. The authors provide an extensive analysis of the combined electron-ion dynamics and explain how to isolate the different dominant mechanism in an otherwise complex intertwining between different degrees of freedom. Although the overall presentation is clear and adequate for a Nature journal some of the evidence for the finding in the paper is staked up in a way that is not always easy to follow (see comments below).

The topic is of interesting and the paper has the potential to represent a theoretical breakthrough to our current understanding of the problem.

However, I have two major points that could undermine the validity of the thesis presented that the authors should address:

- 1. It is well known that, in order to describe excitons with DFT/TDDFT, one needs to choose a functional capable to capture the long range electron-hole interaction and the screening. This is typically achieved by hybrid functionals (e.g. HSE) which contain a fraction of exact-exchange. The authors motivate their choice of a semi-local functional (PBE) by the observation that the system in the CDW phase presents a indirect gap and therefore the screening, which is proportional to $1/q$, is never divergent because the momentum transfer, q , connecting valence and conduction pockets is finite. While this observation is justified for the CDW phase in equilibrium in general this is not the case, in particular during the CDW melting process the system can find itself into direct-gap configuration for which the finite q argument is no longer valid and the xc functional is inadequate. How the authors justify their approximation in this situation?*

Response: We thank the reviewer for recognizing this work as a theoretical breakthrough, and pointing out the important concern. We fully agree with the reviewer that during the CDW melting process the system can find itself into a direct-gap configuration, in which the semi-local XC can not accurately describe the direct exciton. Fortunately, the PLD driving forces are produced only

by the inter-valley excitonic binding, namely as a derivative of total energy with respect to position operator around $\mathbf{q} = \mathbf{w}$. The vertical excitons are important in optical properties especially in the direct-gap configurations during the CDW melting, while it has a limited influence on the PLD dynamics, the major focus of the present work.

In addition, we have performed new calculations using Bethe-Salpeter Equation (BSE) kernels to check on the effects of exciton binding quantitatively. In Fig. S3, we compare the linear-response TDDFT results based on adiabatic PBE (APBE) with those using BSE kernels. One sees that at $\mathbf{q} = \mathbf{w}$, the APBE kernel yields similar spectra with those from the BSE kernel. Since the BSE kernel is a well estimated accurate description of excitons [84, 85], it indicates the electron-hole exchange effect has been adequately described. Therefore, due to the unique band structures of TiSe_2 , the semilocal XC yields acceptable excitonic interactions for the PLD dynamics.

We thank the reviewer for this important question, so we elaborately discuss the validation of our approximations on Page 2 of the revised manuscript as follows:

“It is well known that the semilocal exchange-correlation (XC) functionals (e.g. PBE) poorly describes the long-range Coulomb screening $2\pi/|\mathbf{k}' - \mathbf{k}| \sim \infty$ in vertical excitations $\mathbf{q} = \mathbf{k}' - \mathbf{k} = 0$ [83]. Computationally expensive corrections such as Bethe-Salpeter equation (BSE) [84, 85] can considerably improve the accuracy. As shown in Fig. S2, the experimentally-observed superlinear feature is absent in our simulations. However, the long-range attractions produce spatially uniform forces on the ions. Based on the concept of the excitonic insulator [24–26], the PLD stability is only affected by the inter-valley excitons formed by an attractive interaction $V(\mathbf{w})$ between the electron pocket at the M point and the hole pocket at the Γ point, as shown in Fig. 2(d). Here, $\mathbf{w} = \pm \mathbf{b}_i/2$ and \mathbf{b}_i ($i = 1, 2$) being the reciprocal lattice vector along the i th direction. Therefore, $V(\mathbf{w})$ is a short-range interaction with a characteristic length scale $1/|\mathbf{w}| = a$, where $a = |\mathbf{a}_i|$ and \mathbf{a}_i is the lattice vector. Note $V(\mathbf{w})$ is different from typical long-range interactions in Wannier excitons. The semilocal XC already includes exciton binding between the electron and hole pockets at M and Γ , respectively, albeit slightly underestimating the screening effect.

To quantitatively demonstrate the validity of the semilocal functional in describing the intervalley exciton with momentum $\mathbf{q} = \mathbf{w}$, we compare the linear-response TDDFT results obtained from adiabatic PBE (APBE) and BSE kernels. As shown in Fig. S3, at $\mathbf{q} = \mathbf{w}$, the APBE kernel yields similar absorption spectra with those from BSE kernel. Since the BSE kernel is a well-accepted accurate description of excitons [84, 85], this indicates that the electron-hole exchange effect has been well described in the semilocal XC. Therefore, due to the unique band structures of TiSe_2 , the semilocal XC yields acceptable excitonic interactions.”

Comments: 2. How the experimental ARPES data of fig 5 has been obtained? In the paper it reminds the reader to Ref [65]. However the only figure in the reference that could contain data compatible with the figure is [65] fig. 3 which is a density plot that, at first sight, presents no clear connection with the data points reported in fig 5. The figure in [65] has a much larger time scale up to 2000 fs and the density is so broad that at the very least the presentation of the data in the paper without error bars is misleading.

Since this comparison constitutes the principal validation of the theory on the CDW melting it is

crucial that the authors provide a clear account for the origin of the data.

Response: We thank the reviewer for this good suggestion. To improve the presentation on the comparison between theory and experimental data, we followed the reviewer’s suggestion and replot the Fig. 5 by incorporating the original experimental data with broad intensity distributions from Fig. 3 of Ref. [66]. See the revised Fig. 5. As the reviewer correctly pointed out, although the experimental data has a broader energy resolution and longer timescale, good agreement between theoretical simulations and experimental measurements were found for the first 100 fs when the most interesting features arise.

FIG. L1. (REVISED Fig. 5): Snapshots of TD-EBS for (a-e) the PLD dynamics, and (h-l) the quenched case. EBS at Γ and M point as a function of time for (f-g) PLD dynamics and (m-n) the quenched case. The color bar denotes the carrier population. The squares in (g) mark the experimental tr-ARPES data, reproduced from Ref. [66].

Comments: *In addition I have other minor points that the authors should consider:*

3. *From the text it not clear whether the theoretical result are obtained from a monolayer TiSe_2 or bulk. For instance the data in Fig. 2 present a comparison with experiments on monolayer while Fig. 3 compares to bulk data.*

Response: Sorry for the confusion. We clarify that the bulk TiSe_2 structure is used throughout this work. To be consistent, in the revised manuscript the experimental data in Fig. 2 is changed to the bulk measurements from Ref. [63]. We add more explanation on Page 2 of the revised manuscript as follows:

“The bulk TiSe_2 is used with the interlayer separation of 6.69 Å. ”

Comments: 4. *It is remarkable the fact that the simulations, without any prior knowledge, reproduces the excitation of the out-of-plane A_{1g} phonon mode observed in IR and EELS spectra. The authors find a mode frequency of 4.5 THz while in the literature the value is 3.4 THz (and not 4 THz*

as reported in the paper)[M. Porer, et al. *Nat. Mat.* 13, 857 (2014)]. Can the authors comment on the origin of the mode stiffening predicted by TDDFT?

Response: Thank the reviewer for pointing it out. We have double checked the experimental data accordingly. We would like to clarify that there are in fact two different A_{1g} modes in literature: the out-of-plane A_{1g} phonon at $195\text{-}203\text{ cm}^{-1}$ (5.9 THz) and the A_{1g} CDW amplitude mode at 115 cm^{-1} (3.4 THz) [90, 91]. Both modes are observed, without any prior knowledge, in our TDDFT simulations, as correctly pointed out by the reviewer.

In our simulations (Fig. 4f), the out-of-plane A_{1g} mode has a periodicity of 175 fs (5.7 THz) at low laser fluence ($I = I_0/4$), consistent with the experiment value (5.9 THz). Higher laser fluence $I = I_0$ further decreases the periodicity to 195 fs (5.1 THz), due to the weakened Ti-Se bonds by photocarriers upon strong laser excitation.

The A_{1g} CDW mode has a periodicity of $\sim 220\text{-}240$ fs (4.2-4.5 THz) in our TDDFT simulations (Fig. 4d), which is higher than the experiment (3.4 THz). We attribute this difference to possible mixing of other phonon modes (e.g. E_g phonon, out-of-plane A_{1g} mode, folded optical mode at M-point, etc.) and inadequate accuracy of semilocal functionals to treat low-energy phonons.

In the revised manuscript, we have updated the information and related discussions on Page 3 as follows:

“This is the established out-of-plane A_{1g} mode with the periodicity of 175 fs (5.7 THz) when $I = I_0/4$, which is slightly smaller than the experimental value 5.9 THz [90, 91]. Higher laser fluence $I = I_0$ further decreases the frequency to 5.1 THz, due to the weakened Ti-Se bonds by photocarriers.”

”The calculated double-well PES yields a vibration frequency of 4.3 THz, which is consistent with the oscillation period of the amplitude mode of about 220-240 fs observed in dynamic simulations (Fig. 4d). This frequency is comparable to the experimental value of 3.4 THz observed for the A_{1g} CDW amplitude mode [90, 91]. The difference between theory and experiment is attributed to possible mixing of other phonon modes and inadequate accuracy of semilocal functionals to treat low-energy phonons. The softening of the double-well potential is evidenced by the fact that the CDW mode oscillations disappear when the laser fluence increases from $I_0/4$ to I_0 .”

Comments: 5. In order to highlight the role of the electron-phonon coupling in the lattice distortion the authors perform a simulation with a thermostat that keeps the ion energy fixed. How is the thermostat implemented? Can the authors provide a more details and/or a reference?

Response: In the revised manuscript, we add more detailed descriptions on the dynamics-related algorithms, including forces and thermostat, on Page ID as follows:

“Once the self-consistency in charge density evolution is satisfied, post-processing including the calculation of total energy, Hellmann-Feynman forces, and the ionic trajectory are invoked. For instance, the forces acting on the ions can be calculated through

$$\mathbf{F}_{\mathbf{R}_I} = \sum_{i\mathbf{k}} \langle \psi_{i\mathbf{k}} | \nabla_{\mathbf{R}_I} \mathcal{H} | \psi_{i\mathbf{k}} \rangle, \quad (\text{L1})$$

where \mathbf{R}_I and $\mathbf{F}_{\mathbf{R}_I}$ are the position and force of I th ion.

With \mathbf{R}_I and $\mathbf{F}_{\mathbf{R}_I}$, we utilize the Ehrenfest theorem for evolving ions according to the equation of motion

$$M_I \frac{d^2 \mathbf{R}_I}{dt^2} = \mathbf{F}_{\mathbf{R}_I}, \quad (\text{L2})$$

where M_I is the mass of I th ion. The velocity $v_I(t) = d\mathbf{R}_I/dt$ and the temperature $T(t) = \sum_I^{N_I} M_I v_I^2(t)/2N_I$ are also calculated, where N_I is the total number of ions.

Besides the conventional NVE ensemble, additional thermostats, such as Nosé-Hoover [115, 116] and Berendsen [117] is considered to simulate different environmental conditions. In the damped MD simulations, we utilize a simple velocity-rescaling thermostat. The ionic velocities are rescaled at each time step as

$$\mathbf{v}'_I(t) = \mathbf{v}_I(t) \sqrt{T'(t)/T(t)}, \quad (\text{L3})$$

where $\mathbf{v}'_I(t)$ and $T'(t) = T(t) - \Delta T$ are the rescaled velocity and temperature, respectively. The decreasing rate $\Delta T = 0.01$ eV/atom/ps is used in the simulations.”

Comments: 6. *The meaning of the sentence "Since the PLD dynamics is self-dependent, we infer that there exists a self-amplified electron-phonon mechanism in the PLD dynamics initiated by exciton binding" is a bit obscure. What does it mean that the PLD dynamics is "self-dependent"? Can the authors rephrase to clarify the concept?*

Response: Thank the reviewer for the suggestion. We have rephrased the sentence into:

“ Since the PLD dynamics is sensitively dependent on its own trajectory, we infer that there exists a self-amplified electron-phonon mechanism in the PLD dynamics initiated by exciton binding. ”

Comments: 7. *In Fig. S2 the authors present the photoinduced carrier density. Can the authors provide details how this quantity is calculated?*

It is known that simply projecting the time dependent KS state $\psi_{\mathbf{k}}(t)$ on the the conduction bands at $t=0$ and same \mathbf{k} is not gauge independent and one should project the to equilibrium states at momentum $\mathbf{k}' = \mathbf{k} - \mathbf{A}(t)/c$ - see [T. Otake, et al., Phys Rev B 77, 165104 (2008)] and [S. A. Sato and K. Yabana, Phys Rev B 89, 224305 (2014)]. Failing to do so provides artificial oscillations in the carrier density as a function of the the vector potential $A(t)$ like the ones observed in the first panel of the figure.

Response: Thank the reviewer for the critical insight, to which we are aware of and fully agree. We calculated the carrier density by projecting consistently on \mathbf{k} and observed the oscillations as the reviewer correctly pointed out. When the laser field is present, the carrier density should be projected on the KS state $\psi_{\mathbf{k}'}(t)$ with $\mathbf{k}' = \mathbf{k} - \mathbf{A}(t)/c$. However, when the laser field diminishes $A(t) = 0$, $\mathbf{k}' = \mathbf{k}$ and thus either of the projection methods produce the same results. We clarify that in Fig. S2 the carrier density is all calculated after the laser pulse ends and the $A(t)$ goes to zero.

In the revised text, we have cited [T. Otake, et al., Phys Rev B 77, 165104 (2008)] and [S. A. Sato and K. Yabana, Phys Rev B 89, 224305 (2014)] for the correct description of carrier density projection. More discussion is also added on Page II:

“We investigate the photo-carrier density as a function of time and laser fluence $n(t, I)$. Here, the carrier density is calculated as $n(t) = \frac{1}{2} \sum_{i, \mathbf{k}} |q_{i\mathbf{k}}(t) - q_{i\mathbf{k}}(t=0)|$. We note that $n(t)$ accurately describes the number of excited carriers after the laser field ends $A(t) = 0$. Otherwise, a gauge independent projection on $\psi_{k'}(t)$ with $k' = k - A(t)/c$ instead of $k' = k$ can be used [71–79]. These two projections are identical when the laser field ends $A(t) = 0$.”

Comments: 8. *A remark about point iii) of the PLD dynamics (page 4 left column line 7 from the bottom): “The photoexcited system might relax into the ground state after the recombination of photocarriers, which occurs beyond our simulation period”. While I agree that this would be the expected physical long time behavior I think recombination will never be observable in the numerical model presented in this paper because of the lack of dissipation channels like e.g. electron-electron scattering, electron-phonon scattering, electron-defect scattering etc.*

Response: We thank the reviewer for the comment. As the reviewer correctly pointed out, the recombination requires multiple channels and large-scale simulations to include properly the dissipation and decoherence effects. We add more explanations on Page 14:

“Combining the TDKS equation and the Ehrenfest theorem, the many-body electron-electron interaction and the ionic movement under the excited-state TDKS wavefunction evolution are described in an *ab initio* way. We expect that the electron-electron interactions at the adiabatic XC level and electron-phonon scatterings within the mean-field average trajectory are present in these simulations. The excess electronic energy could dissipate into available phonon modes via electron-phonon coupling or to low-energy electrons via electron-electron scattering, resulting in carrier thermalization and cooling effect.”

and on Page 4:

“iii) The photoexcited system might relax into the ground state after the recombination of photocarriers. This process is beyond our simulations and requires multichannel and large-scale modeling to properly account for the decoherence [92–94] and dissipation effects [95, 96]. Here, we complete the story by briefly discussing the long-time behaviors: in the experiments, the ionic movements in different unit cells have different phases due to the finite thickness of the material, the inhomogeneous spatial distribution of the laser spot, as well as thermal fluctuations. Via phonon-phonon scattering, the equilibrium temperature gradually forms at the timescale of picoseconds. The thermally driven phase transitions may occur through an increasing population of phonons and changes in the atomic potential via anharmonic phonon interactions. Thus, the nonthermal CDW dynamics observed here is separated from thermal transitions at different time scales.”

Reviewer 2

Comments: *Lian et al. present a time-dependent DFT study of CDW ordered TiSe₂. The authors simulate a scenario of femtosecond charge-transfer photoexcitation with high excess energy with parameters adapted to recent experimental studies.*

In their manuscript, the authors report a fast suppression of the charge order on a timescale of 20 fs subsequent to a coherent charge-transfer excitation via an ultrafast field-pulse. The excitation process causes a population of antibonding states that leads to a reduced ordering of the electronic degree of freedom on a timescale of 20 fs. This relaxes the lattice distortion associated with the CDW. The latter process is found to amplify the reduction of the electronic charge ordering ('self-amplification').

The questions addressed in this paper (in particular the role of the crystal lattice in a self-amplified quench of a correlation induced ordering phenomenon) are of interest in the wider field of correlated electron systems and their dynamics on ultrafast timescales.

However, I do not recommend publication of the paper in the present state.

The main issue I see is the insufficient description of how the authors determine the structural dynamics from their calculations and how the DFT results are compared with experimental data. Before publication, the authors need to clarify following issues:

- In their DFT calculation, they set a on-site Coulomb repulsion of the Ti sites of $U_{Ti} = 3.5$ eV. It has been shown earlier [Bianco et al., PRB 92 094107 (2015)] that this approach indeed yields a good between the resulting DFT band structure and ARPES data, but it suppresses the CDW instability. Can the authors show their actual potential energy surface before the field pulse used in their time-dependent calculation sets in (i.e. not only show a schematic picture in Fig. 4(c))? Why is it different to that obtained by Bianco et al.? For low excitation they find an oscillation period of the amplitude mode of about 220 fs (~ 4.5 THz). How does this match with a double well potential for the CDW amplitude mode that is expected to strongly soften (or even become a single well-potential) due to a rather high value of U_{Ti} ?

Response: We would like to express our thanks to the reviewer for recognizing the significance of this work and critical suggestions which help to improve the quality of the manuscript.

First, we clarify that the DFT+U is used only for the band structure calculations in the present work. As correctly pointed out by the reviewer, the PLD stability is maintained with the adoption of PBE, instead of PBE+U. Following the reviewer's suggestion, we plot the potential energy surface (PES) calculated in the ground state with PBE functional in Fig. 4(c). The ground state PES is consistent with the results obtained by Bianco et al. [88]. The calculated double-well PES yields a low-energy vibration frequency of 4.3 THz with the effective mass of 20138 atomic unit, consistent with the oscillation period of the amplitude mode of about 220 fs observed in dynamic simulations. The softening of the double-well potential is evidenced by the fact that the oscillations in the CDW amplitude mode disappear when the laser fluence increases from $I_0/4$ to I_0 (Fig. 4d).

We have explicitly clarified this point and incorporated more discussion on Page 6 as follows:

“Onsite Coulomb repulsion $U = 3.5$ eV was added to the Ti atom to reproduce the

experimental band structure, while we used $U = 0$ in the dynamic TDDFT calculations and structural optimization.”

and on Page 3:

“The calculated double-well PES yields a vibration frequency of 4.3 THz, which is consistent with the oscillation period of the amplitude mode of about 220-240 fs observed in dynamic simulations (Fig. 4d). This frequency is comparable to the experimental value of 3.4 THz observed for the A_{1g} CDW amplitude mode [90, 91]. The difference between theory and experiment is attributed to possible mixing of other phonon modes and inadequate accuracy of semilocal functionals to treat low-energy phonons. The softening of the double-well potential is evidenced by the fact that the CDW mode oscillations disappear when the laser fluence increases from $I_0/4$ to I_0 .”

Comments: - *It is unclear to me from the manuscript how the structural motions are obtained from the potential. What is the equation of motion and which effective mass is used for the modes? How are the mode frequencies in Fig. 4 (CDW amplitude mode, out-of-plane mode) obtained?*

Response: We thank the reviewer for raising these questions. The structural motions are simulated *ab initio* without assuming any empirical parameters (such as the shape of potential, effective mass etc.). The *ab initio* simulations were carried out based on two fundamental theorems:

1. Runge-Gross theorem in the form of TDKS equations, describing the dynamic density-potential mapping and the evolution of electrons;
2. Ehrenfest theorem, which gives the forces on the ions.

Thus, instead of moving the structure based on *a priori* potential energy surface and effective mass, in the current approach the PES for all ions is generated on-the-fly via *ab initio* calculations along the trajectory. The result would be the same as that assuming *a priori* PES (similar to the calculated Fig. 4c) and the effective mass (20138 a.u.). The equation of motion for ions is Newtonian dynamics but on the excited state PES.

The out-of-plane motions are directly observed from the trajectory of *ab initio* TDDFT-MD simulations, shown in Fig. 4. The frequencies of the phonon mode are obtained as follows:

1. The periodicities of the oscillation are read from the figure, which are 195 fs for laser fluence I_0 and 175 fs for $I_0/4$.
2. Convert the timescales to frequencies via the equation $f = 1000/t_0$, where f and t_0 are in units of THz and fs respectively. This generates the values of 5.1 THz for I_0 and 5.7 THz for $I_0/4$.

To clarify, we add more details of simulation algorithms on how the forces and dynamics are calculated on Page 13:

“Once the self-consistency in charge density evolution is satisfied, post-processing including the calculation of total energy, Hellmann-Feynman forces, and the ionic trajec-

tory are invoked. For instance, the forces acting on the ions can be calculated through

$$\mathbf{F}_{\mathbf{R}_I} = \sum_{i\mathbf{k}} \langle \psi_{i\mathbf{k}} | \nabla_{\mathbf{R}_I} \mathcal{H} | \psi_{i\mathbf{k}} \rangle, \quad (\text{L4})$$

where \mathbf{R}_I and $\mathbf{F}_{\mathbf{R}_I}$ are the position and force of I th ion.

With \mathbf{R}_I and $\mathbf{F}_{\mathbf{R}_I}$, we utilize the Ehrenfest theorem for evolving ions according to the equation of motion

$$M_I \frac{d^2 \mathbf{R}_I}{dt^2} = \mathbf{F}_{\mathbf{R}_I}, \quad (\text{L5})$$

where M_I is the mass of I th ion. The velocity $v_I(t) = d\mathbf{R}_I/dt$ and the temperature $T(t) = \sum_I^{N_I} M_I v_I^2(t) / 2N_I$ are also calculated, where N_I is the total number of ions.

Besides the conventional NVE ensemble, additional thermostats, such as Nosé-Hoover [115, 116] and Berendsen [117] is considered to simulate different environmental conditions. In the damped MD simulations, we utilize a simple velocity-rescaling thermostat. The ionic velocities are rescaled at each time step as

$$\mathbf{v}'_I(t) = \mathbf{v}_I(t) \sqrt{T'(t)/T(t)}, \quad (\text{L6})$$

where $\mathbf{v}'_I(t)$ and $T'(t) = T(t) - \Delta T$ are the rescaled velocity and temperature, respectively. The decreasing rate $\Delta T = 0.01$ eV/atom/ps is used in the simulations.”

In addition, we present the calculated potential energy surface in Fig. 4(c) and discuss the consistency between the PES and the frequencies obtained from dynamic simulations on Page 3.

Comments: - *How is the dissipation of electronic excess energy or carrier cooling implemented (Fig 5 (a-e))? Is the electronic system only cooled via manual dissipation of energy solely via the coupled A_{1g} coherent mode?*

Response: We thank the reviewer for raising this question. As we emphasized above, there is no manual dissipation channel added, since all the electron-electron and electron-phonon couplings are automatically taken into consideration in *ab initio* simulations given an accurate exchange-correlation functional. The dissipation of electronic excess energy (carrier cooling) is achieved by energy transfer into all possible phonon modes via electron-phonon couplings and to thermalized low-energy electrons via carrier multiplication processes.

We explain the algorithm used in our simulations on Page 14 as follows:

“Combining the TDKS equation and the Ehrenfest theorem, the many-body electron-electron interaction and the ionic movement under the excited-state TDKS wavefunction evolution are described in an *ab initio* way. We expect that the electron-electron interactions at the adiabatic XC level and electron-phonon scatterings within the mean-field average trajectory are present in these simulations. The excess electronic energy could dissipate into available phonon modes via electron-phonon coupling or to low-energy electrons via electron-electron scattering, resulting in carrier thermalization and cooling effect.”

Comments: *Further comments: The authors should elaborate why they claim that their results*

support excitonic interactions. In my opinion the results could equally well be interpreted in terms of the Jahn-Teller effect.

Response: We have incorporated more elaborate discussions in the revised manuscript as follows:

“There are open discussions in literature on whether excitonic interaction or the Jahn-Teller distortion is dominant for the formation of CDW in TiSe₂. Our simulations seem to indicate the two effects could coexist in a short period perturbed by photoexcitation. From Fig. 3 and 4 it is shown that while there is no significant lattice distortion (thus no changes in Jahn-Teller interaction), the electronic order is suppressed by 20% in the first 20 fs, beaconing excitonic interactions. On the other hand, the following self-amplifying process suggests dynamic Jahn-Teller lattice distortion is crucial for the subsequent decay/formation of CDW state.”

Comments: - *Fig. 2 (d) does not show the meaning of $V(w)$ as written in the text.*

Response: Thank the reviewer for the good suggestion. Now we show the meaning of $V(w)$ in the revised Fig. 2.

Comments: - *I am wondering to which extent the link between a 'nonthermal' phase transition (ref. 64) and the kinetic energy of lattice sites during the ultrafast transition is justified. Nonthermal in the sense of ref. 64 refers to the driving mechanism, meaning that the structural soft-mode potential is relaxed or driven to a single well-potential by excitation of the electronic degree of freedom (i.e. direct coupling of the structural potential to the electronic state). For a thermally driven phase transition, an increasing population of the vibrational system (e.g. via e-ph scattering after laser excitation) would change the structural potential via anharmonic phonon interactions. In this case the kinetic energy of the respective ions would be even lower as the structural dynamics induced by a changing potential is limited by the timescale of e-ph heat exchange.*

Response: We thank the reviewer for the comment. The reviewer is correct that the low kinetic energy may not be directly connected to nonthermal processes. The two mechanisms, thermal and nonthermal, however differs in the timescales they exist and whether excited-state carriers are present. Since all the three steps discussed in the manuscript are short-lived (<300 fs) and the excited-state carriers are clearly present (Fig. 5), we tend to think the processes discussed is primarily nonthermal.

We have modified the relevant sentences and added more discussions in the revised text:

“Although the low kinetic energy of ions may not be directly connected to nonthermal processes, we believe the processes discussed above are primarily nonthermal since they are short-lived (<300 fs) and the excited-state carriers are clearly present (Fig. 5). In this case the structural soft-mode potential is relaxed or even driven to a single well-potential by excitation of the electronic degree of freedom. This is also consistent with the experimental observations [65] and previous studies [97–99]. ”

and

“iii) The photoexcited system might relax into the ground state after the recombination of photocarriers. This process is beyond our simulations and requires multichannel and

large-scale modeling to properly account for the decoherence [92–94] and dissipation effects [95, 96]. Here, we complete the story by briefly discussing the long-time behaviors: in the experiments, the ionic movements in different unit cells have different phases due to the finite thickness of the material, the inhomogeneous spatial distribution of the laser spot, as well as thermal fluctuations. Via phonon-phonon scattering, the equilibrium temperature gradually forms at the timescale of picoseconds. The thermally driven phase transitions may occur through an increasing population of phonons and changes in the atomic potential via anharmonic phonon interactions. Thus, the nonthermal CDW dynamics observed here is separated from thermal transitions at different time scales.”

Comments: *I recommend an extended revision of the manuscript that includes addressing the above points. Furthermore, I suggest publication in a more specialised journal as the manuscript reports results obtained by an established computational technique for a specific class of materials.*

Response: We thank the reviewer for the comments. We respectfully disagree with the statement that this manuscript ”reports results obtained by an established computational technique for a specific class of materials”. We emphasize that the computational methodology developed here is new with several important innovations rendering this investigation possible, which is well distinguished from the established methods.

i) We can directly include photons, phonons, and electrons on the same footing in the simulations, by incorporating all into the Hamiltonian using gauge-field theory, to describe the full quantum mechanical dynamics beyond the perturbative regime.

ii) Utilizing the new efficient algorithm, we can integrate the TDKS equation with a 0.2 fs timestep, which is over 200 times faster than regular TDDFT packages such as OCTOPUS and ELK. This paves the way to simulate the whole dynamics ranging from sub-fs to 500 fs. We also made intensive efforts to include the PAW potential in our TDDFT implementation, which enables the state-of-the-art description of most accurate electron-ion interactions.

iii) Finally, we successfully make use of the band unfolding techniques in dynamic TDDFT simulations to reproduce the effective band structure over the full Brillouin zone, which can be directly compared with the experimental time-resolved angle-resolved photoemission spectroscopy data (tr-ARPES). Such a comparison on the *ab initio* level was not demonstrated before.

In summary, the techniques first developed here have clear novelty and advantages compared to the established methods. As we explained in the revised manuscript, the algorithm is general and full *ab initio*. Without *a priori* equations and parameters, this method is not designed for a specific class of materials; rather, it is readily applied to all kind of materials of interest.

Comments: *The authors present a real-time TDDFT study of photoexcited carrier dynamics in the material 1T-TiSe₂, investigating directly from first-principles, the interplay between charge-density-wave (CDW) order and periodic lattice distortions (PLD) in the time-domain. Starting from a ground state where CDW order and PLD co-exist, the authors show that photoexcitation reduces CDW order at early times (< 20 fs) resulting in excited-state forces that in turn drive a switching of PLD through nonadiabatic electron-ion dynamics on the 20-300 fs timescale. The simulations in this work certainly present a detailed atomistic picture of photoexcitation dynamics in 1T-TiSe₂ and a plausible interpretation of some time-resolved ARPES experiments published previously. The methods employed by the authors namely real-time TDDFT combined with Ehrenfest dynamics are state-of-the art in a condensed-matter context. The simulations are also complemented by additional post-processing analysis which serve to clarify the detailed dynamics while making contact with ultrafast experiments.*

Overall this is a detailed and impressive set of simulations and I would recommend publication after the authors have addressed the concerns/questions below:

Comments: *(1) On page 2, the authors state: Thus, the semilocal XC and electron-ion dynamics simulations are suitable for tracking photoexcitation physics of TiSe₂.*

This statement needs some qualification. The argument preceding the above statement only justifies the use of semi-local functionals for an excitonic coupling that is short-ranged in real-space. It is not clear whether all aspects of photoexcitation including direct inter-band excitations and related excitonic effects are adequately captured at the level of semi-local functionals. Indeed in Fig. S2, the authors show that experimental values for photoexcitation differ from the simulations at intermediate intensities and attribute the difference to excitonic effects.

Response: We are grateful for the strong supports and pertinent comments from the reviewer on the techniques and major findings of this work. Below, we address the reviewer's good suggestions.

We fully agree with the reviewer that more qualifications on the semilocal XC to describe exciton interactions are needed. Following his/her good suggestion, we performed new calculations using linear-response TDDFT and compared directly the results from semilocal adiabatic PBE kernel and those from more sophisticated Bether-Salpeter-Equation (BSE) kernel, as presented in Fig. S3. We found that, as we expected, both APBE and BSE kernels yield similar absorption spectra, suggesting that the electron-hole exchange effect is not critical for the entangled electron-ion dynamics in TiSe₂. As the reviewer pointed out, Fig. S2 shows some difference with the experimental value, which comes from regular excitons excited vertically in the \mathbf{k} space. This effect might be critical for investigating the optical properties, such as reproducing the superlinear behaviors in photoabsorption, but it would barely affect the PLD dynamics.

We have incorporated above discussions into the revised text on Page 2:

“It is well known that the semilocal exchange-correlation (XC) functionals (e.g. PBE) poorly describes the long-range Coulomb screening $2\pi/|\mathbf{k}' - \mathbf{k}| \sim \infty$ in vertical excitations $\mathbf{q} = \mathbf{k}' - \mathbf{k} = 0$ [83]. Computationally expensive corrections such as Bethe-Salpeter equation (BSE) [84, 85] can considerably improve the accuracy. As shown in Fig. S2,

the experimentally-observed superlinear feature is absent in our simulations. However, the long-range attractions produce spatially uniform forces on the ions. Based on the concept of the excitonic insulator [24–26], the PLD stability is only affected by the inter-valley excitons formed by an attractive interaction $V(\mathbf{w})$ between the electron pocket at the M point and the hole pocket at the Γ point, as shown in Fig. 2(d). Here, $\mathbf{w} = \pm \mathbf{b}_i/2$ and \mathbf{b}_i ($i = 1, 2$) being the reciprocal lattice vector along the i th direction. Therefore, $V(\mathbf{w})$ is a short-range interaction with a characteristic length scale $1/|\mathbf{w}| = a$, where $a = |\mathbf{a}_i|$ and \mathbf{a}_i is the lattice vector. Note $V(\mathbf{w})$ is different from typical long-range interactions in Wannier excitons. The semilocal XC already includes exciton binding between the electron and hole pockets at M and Γ , respectively, albeit slightly underestimating the screening effect.

To quantitatively demonstrate the validity of the semilocal functional in describing the intervalley exciton with momentum $\mathbf{q} = \mathbf{w}$, we compare the linear-response TDDFT results obtained from adiabatic PBE (APBE) and BSE kernels. As shown in Fig. S3, at $\mathbf{q} = \mathbf{w}$, the APBE kernel yields similar absorption spectra with those from BSE kernel. Since the BSE kernel is a well-accepted accurate description of excitons [84, 85], this indicates that the electron-hole exchange effect has been well described in the semilocal XC. Therefore, due to the unique band structures of TiSe_2 , the semilocal XC yields acceptable excitonic interactions.”

Comments: (2) On page 4, the authors state: In contrast to the complete PLD inversion for the $I = I_0$ case, $\mathbf{d}_i(t)$ only oscillates around the original value when laser intensity decreases to $I_0/4$.

What is the reason for this type of intensity dependence? This suggests that the amplitude of the ionic motions perhaps depends on the density of excited carriers rather than on the energy of excitation. Is this an artifact of employing the Ehrenfest approximation in calculating nuclear forces? Is there experimental evidence indicating that the PLD switch would exhibit a fluence dependence as the simulations suggest?

Response: We thank the reviewer for noticing this important question. We would like to clarify that, as the reviewer correctly pointed out, the PLD dynamics is sensitive to the carrier density instead of photon energy. In its undistorted phase, 1T- TiSe_2 is a semimetal; while there emerges a small band gap of 0.18 eV when it enters the CDW phase. In both experimental studies and our simulations, the photon energy is large enough to pump the electrons from the valence bands into the conduction bands. This insensitive dependence on photon energy is also experimentally observed: in Ref. [65], the authors utilize two different photon energies, 1.55 eV (800 nm) and 0.78 eV (1580 nm), to excite the material and obtain consistent lattice dynamics under similar intensities (Fig. 1 and 2).

In contrast, carrier density is closely related to the extent of chemical bonding. Excited electrons are generally pumped from the bonding states to the anti-bonding states, which change the potential energy surface, as demonstrated in Fig. 4(c). Thus, higher laser fluence and larger carrier density substantially weaken the chemical bonds and soften the phonon mode. Interestingly, this fluence dependence is also observed in Ref. [65] [Fig. 1(c)] and Ref. [67] [Fig. 4]: the A_{1g} CDW mode is softened as the laser fluence increases, which serves as a precursor of the PLD melting.

We conclude that the intensity-dependence of the PLD dynamics is not an artifact of the Ehrenfest algorithm. In the revised manuscript, we have explicitly discussed the dependence of ultrafast

dynamics on fluence and photon energy on Page 16:

“Consistent with experimental observations [65], we note that the PLD dynamics are not sensitive to the photon energies. This is because that the band gap of CDW 1T-TiSe₂ (0.18 eV) is smaller than the photon energy in most commonly used laser sources (~ 1 eV). Besides, the sub-picosecond laser pulses utilized in the experiments and our simulations would bring up significant broadening in photon energy as well as multi-photon absorption processes.”

and on Page 3:

“As shown in Fig. 4(c), we generate the dynamical PES by plotting $-E_{\text{kin}}(t)$ as a function of $\delta(t)$. This implies a scenario of ultrafast PLD switch, which includes three consecutive steps: i) In the first 20 fs, the laser pulses pump the electrons from the bonding state to the anti-bonding state, which leads to the change of the potential energy surface....”

“This intensity dependence is also experimentally observed [65, 67]: the A_{1g} CDW mode is softened as the laser fluence increases, which serves as a precursor of the PLD melting.”

Comments: (3) *The Methodology section should provide details of the type of electron-ion dynamics employed in the study including parameters related to the thermostat used and relevant literature citations in this regard.*

Response: Following the reviewer’s instruction, we update the description on the methodology as follows:

“Once the self-consistency in charge density evolution is satisfied, post-processing including the calculation of total energy, Hellmann-Feynman forces, and the ionic trajectory are invoked. For instance, the forces acting on the ions can be calculated through

$$\mathbf{F}_{\mathbf{R}_I} = \sum_{i\mathbf{k}} \langle \psi_{i\mathbf{k}} | \nabla_{\mathbf{R}_I} \mathcal{H} | \psi_{i\mathbf{k}} \rangle, \quad (\text{L7})$$

where \mathbf{R}_I and $\mathbf{F}_{\mathbf{R}_I}$ are the position and force of I th ion.

With \mathbf{R}_I and $\mathbf{F}_{\mathbf{R}_I}$, we utilize the Ehrenfest theorem for evolving ions according to the equation of motion

$$M_I \frac{d^2 \mathbf{R}_I}{dt^2} = \mathbf{F}_{\mathbf{R}_I}, \quad (\text{L8})$$

where M_I is the mass of I th ion. The velocity $v_I(t) = d\mathbf{R}_I/dt$ and the temperature $T(t) = \sum_I^{N_I} M_I v_I^2(t) / 2N_I$ are also calculated, where N_I is the total number of ions.

Besides the conventional NVE ensemble, additional thermostats, such as Nosé-Hoover [115, 116] and Berendsen [117] is considered to simulate different environmental conditions. In the damped MD simulations, we utilize a simple velocity-rescaling thermostat. The ionic velocities are rescaled at each time step as

$$\mathbf{v}'_I(t) = \mathbf{v}_I(t) \sqrt{T'(t)/T(t)}, \quad (\text{L9})$$

where $v'_I(t)$ and $T'(t) = T(t) - \Delta T$ are the rescaled velocity and temperature, respectively. The decreasing rate $\Delta T = 0.01$ eV/atom/ps is used in the simulations.”

REVIEWERS' COMMENTS:

Reviewer #1 (Remarks to the Author):

The authors' reply clarified the two major points I had with the previous version of the manuscript providing further numerical simulations to support their claims.

They also replied to all the other points I raised in my review to my satisfaction and amended the text accordingly.

I have no further reserve and I recommend the paper for publication.

Reviewer #2 (Remarks to the Author):

Liang et al. addressed each of my points in great detail and improved the manuscript in most of the aspects raised all referees. In their revised manuscript, they sufficiently addressed following issues raised by me:

-The authors now explain in detail how they obtain the structural dynamics, making it clearer for non-expert readers.

-The explanation on how the thermostat is implemented is significantly more detailed now and provides the necessary background information.

-The authors improved the discussion on the connection between kinetic energy of ions and nonthermal processes. Together with the added paragraph I consider their argumentation as valid.

Nonetheless, I still see one issue which I strongly recommend to be addressed before publication: The authors now state that they used a value of $U_{Ti}=0$ in their dynamics calculation while they employ $U_{Ti}=3.5$ to reproduce the experimental band structure. When doing so, it is necessary, in my opinion, to make it clear to the reader that for $U_{Ti}=3.5$ there is no CDW (with respect to purely the charge distribution) without manually fixing the lattice to the CDW phase. Furthermore, I recommend citing the Bianco paper. Since the manuscript mainly focuses on the dynamics of the electronic and structural degrees of freedom, I think it is ultimately not too much of an issue if the PES bandstructure is not matched exactly (with $U_{Ti}=0$), but the authors should provide quantitative background to the reader e.g. by showing their bandstructure for $U_{Ti}=0$ in Fig. 2(e) for comparison.

Finally I suggest publication of the paper by Liang et al. in Nature Communications if the authors follow my recommendations above.

Reviewer #3 (Remarks to the Author):

The authors have satisfactorily addressed my questions and concerns from the first round of reviews through some additional simulations and clarifications in the text. On the basis of the improvements carried out to the manuscript, I would recommend this article for publication.

Reply to reviewer's suggestions

Reviewer 1

Comments: The authors' reply clarified the two major points I had with the previous version of the manuscript providing further numerical simulations to support their claims. They also replied to all the other points I raised in my review to my satisfaction and amended the text accordingly. I have no further reserve and I recommend the paper for publication.

Response: We thank the referee for recommending our paper for publication.

Reviewer 2

Comments: Liang et al. addressed each of my points in great detail and improved the manuscript in most of the aspects raised all referees. In their revised manuscript, they sufficiently addressed following issues raised by me:-The authors now explain in detail how they obtain the structural dynamics, making it clearer for non-expert readers.-The explanation on how the thermostat is implemented is significantly more detailed now and provides the necessary background information.-The authors improved the discussion on the connection between kinetic energy of ions and nonthermal processes. Together with the added paragraph I consider their argumentation as valid.

Nonetheless, I still see one issue which I strongly recommend to be addressed before publication: The authors now state that they used a value of $U_{\text{Ti}}=0$ in their dynamics calculation while they employ $U_{\text{Ti}}=3.5$ to reproduce the experimental band structure. When doing so, it is necessary, in my opinion, to make it clear to the reader that for $U_{\text{Ti}}=3.5$ there is no CDW (with respect to purely the charge distribution) without manually fixing the lattice to the CDW phase. Furthermore, I recommend citing the Bianco paper. Since the manuscript mainly focuses on the dynamics of the electronic and structural degrees of freedom, I think it is ultimately not too much of an issue if the PES bandstructure is not matched exactly (with $U_{\text{Ti}}=0$), but the authors should provide quantitative background to the reader e.g. by showing their bandstructure for $U_{\text{Ti}}=0$ in Fig. 2(e) for comparison.

Finally I suggest publication of the paper by Liang et al. in Nature Communications if the authors follow my recommendations above.

Response: We thank the referee for recommending our paper. We have now added the band structure for $U_{\text{Ti}} = 0$ eV in Fig. 2(e) for comparison. We also added a sentence to clarify that for $U_{\text{Ti}} = 3.5$ eV there is no CDW ground state, and we cited Bianco's paper as the following on Page 2:

“We note that the PBE+U approach with the Hubbard $U_{\text{Ti}} = 3.5$ eV incorrectly predicts that the CDW state is unstable [88]. Thus, we use $U_{\text{Ti}} = 0$ eV for all the dynamic simulations.”

Reviewer 3

Comments: The authors have satisfactorily addressed my questions and concerns from the first round of reviews through some additional simulations and clarifications in the text. On the basis of the improvements carried out to the manuscript, I would recommend this article for publication.

Response: We thank the referee for recommending our paper for publication.